# Adipose Tissue Immunomodulation and Treg/Th17 Imbalance in the Impaired Glucose Metabolism of Children with Obesity

**DOI:** 10.3390/children8070554

**Published:** 2021-06-27

**Authors:** Stefania Croce, Maria Antonietta Avanzini, Corrado Regalbuto, Erika Cordaro, Federica Vinci, Gianvincenzo Zuccotti, Valeria Calcaterra

**Affiliations:** 1Immunology and Transplantation Laboratory, Pediatric Hematology Oncology Unit, Cell Factory, Department of Maternal and Children’s Health, Fondazione IRCCS Policlinico S. Matteo, 27100 Pavia, Italy; stefania_croce186@yahoo.it (S.C.); erika.cordaro01@universitadipavia.it (E.C.); 2Pediatric Unit, Department of Maternal and Children’s Health, Fondazione IRCCS Policlinico S. Matteo and University of Pavia, 27100 Pavia, Italy; corrado.regalbuto01@universitadipavia.it (C.R.); fede90vinci@gmail.com (F.V.); 3Department of Biomedical and Clinical Science, “Luigi Sacco”, University of Milan, 20157 Milan, Italy; gianvincenzo.zuccotti@unimi.it; 4Pediatric Department, “Vittore Buzzi” Children’s Hospital, 20154 Milano, Italy; 5Pediatrics and Adolescentology Unit, Department of Internal Medicine, University of Pavia, 27100 Pavia, Italy; valeria.calcaterra@unipv.it

**Keywords:** childhood obesity, adipose tissue-associated inflammation, Th17, Treg, glucose metabolism disorders

## Abstract

In the last few decades, obesity has increased dramatically in pediatric patients. Obesity is a chronic disease correlated with systemic inflammation, characterized by the presence of CD4 and CD8 T cell infiltration and modified immune response, which contributes to the development of obesity related diseases and metabolic disorders, including impaired glucose metabolism. In particular, Treg and Th17 cells are dynamically balanced under healthy conditions, but imbalance occurs in inflammatory and pathological states, such as obesity. Some studies demonstrated that peripheral Treg and Th17 cells exhibit increased imbalance with worsening of glucose metabolic dysfunction, already in children with obesity. In this review, we considered the role of adipose tissue immunomodulation and the potential role played by Treg/T17 imbalance on the impaired glucose metabolism in pediatric obesity. In the patient care, immune monitoring could play an important role to define preventive strategies of pediatric metabolic disease treatments.

## 1. Introduction

In the last few decades, obesity has dramatically increased in pediatric patients and the link between obesity-induced inflammation and its complications has been described in numerous studies [1,2]. As reported by World Health Organization, the global prevalence of overweight and obesity in children and adolescents aged 5–19 has risen from 4% in 1975 to 18% in 2016 [3]. In 2016 more than 340 million children and adolescents worldwide were in a condition of excess body weight [3].

Obesity is a chronic disease correlated with various factors such as environment, heredity, lifestyle and others [4]. The underlying process is triggered by imbalanced energy intake and consumption [4]. It is well know that systemic inflammation correlates to obesity, characterized by the presence of CD4 and CD8 T cell infiltration and modified immune response, which contributes to the development of obesity related diseases and metabolic disorders like dyslipidemia, type 2 diabetes (T2DM), and cardiovascular pathologies already in pediatric age [5,6].

Adipose tissue (AT) furnishes the organism with a storage of nutrients that is drained during starvation. It produces signals which limit immune cell amount and activity under conditions of nutrient deficiency, allowing proper immune system activity when food sources are available [7]. When excessive AT deposition occurs, it becomes the site of pathological immune system activation, leading to chronic low-grade systemic inflammation. It becomes clear hence that obesity is associated with various disorders in which the immune system plays a key role [8,9]. Naïve T cells are normally quiescent and metabolically inactive, but after stimulation they proliferate and differentiate into various T helper cells (Th), including Th17, and T-regulatory (Treg) cells. Th17 cells primarily fight against extracellular microbial pathogens and mediate autoimmune disease, but they are also known to be involved in allograft rejection. Treg lymphocytes have an opposite function compared to Th17; they attend in modifying the immune response, in order to sustain immune self-tolerance, and prevent autoimmune disease [10,11,12]. Treg produce and secrete, among the others, IL-10 and TGF-β, which could regulate the differentiation and proliferation of lymphocytes and other immune cells [13] suppressing the activation of the immune system. A lack of Treg leads to autoimmune disorders, and a high ratio of Treg/Th17 is proved to be associated with cancer incidence [14,15]. Treg and Th17 cells are dynamically balanced under healthy conditions, but imbalance occurs in inflammatory and pathological states such as obesity [16].

Some studies demonstrated that peripheral Treg and Th17 cells exhibit increased imbalance with worsening of glucose metabolic dysfunction in obese adult and pediatric patients [16,17]. In this narrative review, we considered the role of AT immunomodulation and the potential role played by Treg/Th17 imbalance on the impaired glucose metabolism in pediatric obesity.

## 2. Methods

The aim of this narrative review is to investigate the immunomodulation properties of AT and the role of Treg/Th17 imbalance on the glucometabolic disorders in overweight and obese children. To achieve this, each author independently, identified the most relevant original scientific papers, clinical trials, meta-analyses and reviews in English language, published in the last 15 years. Case reports or series and letters were excluded.

The following keywords were used to search for papers published in the last 15 years in each author’s field of expertise: childhood obesity, pediatric obesity; children metabolic status; adipose tissue-associated inflammation, Th17; Treg; immune system and obesity-related glucose metabolism disorders. Electronic databases (PubMed, Scopus, EMBASE, and Web of Science) were used in the research. The contributions were critically reviewed and collected. The resulting draft was discussed with all co-authors. The final version was approved by all co-authors. As a narrative review, several statements based on expert opinions and not evidence-based or supported by appropriate in vitro or in vivo studies were included.

## 3. Adipose Tissue Immunomodulation

AT is a functionally pleiotropic tissue of mesodermal origin [18]. It is a type of loose connective tissue in which the adipose cells (adipocytes) organize themselves into lobules. It covers different functions, in particular: (i) represents the main reserve of energetic material, performing an important trophic action; (ii) avoids the dispersion of body heat through the skin; (iii) exerts a protective and supportive action, contributing to determine the profile of same tissues/organs. According to the body’s location, AT can be classified as: adipose covering tissue, present at subcutaneous level, representing about 50% of the total AT; internal AT, heterogeneously distributed in the abdominal cavity, being about 45% of the total AT, and muscle infiltration fat, which represents the remaining 5% with the function of assisting the normal muscle performance.

At a functional level, AT could be divided into deposit tissue, that depend to the nutritional state, and support tissue, that never completely disappears. In mammals, AT is differentiated into unilocular AT, commonly known as white fat, critical for energy storage, endocrine communication, and insulin sensitivity and multilocular AT, also called brown fat, which uses energy for non-shivering heat production and is critical for body temperature maintenance [19,20,21,22,23]. Adipocytes are surrounded by connective tissue that includes macrophages, fibroblasts, preadipocytes, and various other cell types found in stromavascular tissue [23,24,25]. The traditional view of AT as a passive reservoir for energy storage is no longer valid. AT expresses and secretes a variety of bioactive peptides, known as adipokines, which act at local (autocrine/paracrine) and systemic (endocrine) level [26] as well as on immune cells [27]. Studies in literature show that AT plays an immune role due to the presence of immune cells residing in the tissue itself, such as macrophages, mast cells, neutrophils, T and B lymphocytes, which are the second most represented cytotype after adipocytes [28], Figure 1. The presence of these cells makes AT an immune organ.

Obesity is well known to be associated with chronic low-grade systemic inflammation, well-defined by modifications of circulating cytokines and acute phase proteins/reactants. Adipokines are involved in the regulation of energy consumption, insulin sensitivity, endothelial function, glucose and lipid metabolism [29].

The mechanisms triggering the inflammatory process at the level of AT are not yet known. In particular, when the caloric intake is higher than daily needs (positive energy balance), the AT undergoes morphological and metabolic changes leading to the release of pro-inflammatory cytokines such as interleukin 6 (IL-6) and tumor necrosis alpha (TNF-α), mediated by the leptin hormone [30]. These conditions are associated with a chronic inflammatory response characterized by abnormal cytokine production, increased acute-phase reactants, and activation of inflammatory signalling pathways [31].

AT hypoxia represents one of the most important factors leading to AT dysfunction along with endoplasmic reticulum stress and oxidative stress as seen in animal and human models [32,33,34]. The excess of nutrients determines local high accumulation of fatty acids with hypertrophic and hyperplastic AT modifications [35]. When tissue mass expands, clusters of adipocytes start to agglomerate distant from the vasculature, without increased proportion of the cardiac output and the extent of the blood flow at the AT level. Adipocytes could be larger than the normal diffusion distance of O_2_ of 100–200 micron and it has been noted that in some tissues the levels of partial pressure of oxygen may be close to zero at only 100 micron from the vasculature [36,37].

Consequently, the cellular and tissue vasculature becomes insufficient with reduced oxygen diffusion within cells, creating a hypoxic state. Immunohistochemical staining of tissue sections demonstrated that hypoxic areas within white AT were colonized by macrophages, suggesting the presence of a clear correlation between hypoxia and the inflammatory state in AT [33]. Obesity-related inflammatory process especially results in the recruitment of neutrophils and pro-inflammatory T cells (Th1 cells) [38]. Th1 cells secrete pro-inflammatory cytokines that stimulate monocyte differentiation into the pro-inflammatory macrophage subtype [31].

### 3.1. Innate Immune System Cells in Adipose Tissue

The innate immune system cells are monocytes/macrophages as well as other myeloid cells including dendritic cells, mast cells, NK cells and granulocytes (neutrophils, basophils, eosinophils), Figure 1.

A critical role for AT-resident immune cells in the regulation of local and systemic metabolic homeostasis has been supposed, with a fine regulated crosstalk between CD4+ T cells and local antigen-presenting cells (APCs), such as macrophages and dendritic cells (DCs). The inflammatory response is characterized by a complex cascade of both pro-inflammatory and anti-inflammatory molecules. The primary immune response, where macrophages play a key role in cooperating with the toll-like receptor family, especially TLR4, start the inflammatory response pathway [39]. Many important molecules are produced during these processes, such as nuclear factor kB (NFkB), pro-inflammatory cytokines like IL-1, IL-6, and TNF-α, serum amyloid A3 (SAA3), α l-acid glycoprotein, the lipocalin 24p3 and plasminogen activator inhibitor-1 (PAI-1).

It is well known that AT macrophages play key roles in the development of the chronic inflammatory state along with metabolic dysfunctions. [1,40,41]. They represent the predominant immune cells in AT, being the main source of inflammatory cytokines, such as TNF-α, IL-6 and IL-1β. An increase in circulating levels of these macrophage-derived factors in obesity leads to a chronic low-grade inflammatory state that has been linked to obesity complications [42].

Macrophages can be divided in two subtypes, differing in membrane antigen expression and secreted cytokines: M1, classically activated macrophages CD11b(+), F4/80(+) and CD206(-) with pro-inflammatory role and M2, alternatively activated macrophages CD11b(+), F4/80(+) and CD206(+) induced by anti-inflammatory molecules (IL-13, IL-10, IL-4 and macrophage colony-stimulating factor) [43].

M1 macrophages take part as inducer and effector cells in polarized Th1 responses, mediating resistance against intracellular parasites and tumoral cells, while M2 express high levels of scavenge and galactose-type receptors, contributing to tissue remodeling, angiogenesis and tumor progression [44,45]. In AT, adiponectin deficiency contributes to the shift of macrophages from the anti-inflammatory M2 to the pro-inflammatory M1 phenotype. Through the secretion of pro-inflammatory cytokines, M1 macrophages will attract peripheral monocytes, which in turn will be polarized into M1 macrophages [46]. The different polarization of macrophages in human AT is dependent on nutritional status, prevailing the M2 phenotype in normal-weight, and the M1 phenotype in overweight-obese children. Correction of the M1/M2 ratio is important to improve their function [47]. Zughe et al. investigating the effect of linagliptin, an inhibitor of dipeptidyl peptidase 4 (DPP-4), observed a macrophage migration and polarization in white AT of HFD-induced obese mice. The study showed a greater number of DPP-4+ macrophages in obese mice than in lean mice. Furthermore, it emerged that linagliptin attenuates oxidative stress, inflammation, and insulin resistance (IR), through reduction of M1-polarized macrophage accumulation and induction of an M2-dominant shift in AT alleviating IR and inflammation in obesity [48].

Recently, it has been demonstrated in mouse models that an accumulation of cholesterol by macrophage, led to the atherogenesis, a typical cardiovascular problem related to obesity [49].

Ying et al. described that in an obese insulin resistant mouse model, AT macrophages (ATMs) secrete miRNA-containing exosomes (Exos). The administration of obese mice-Exos cause glucose intolerance and insulin resistance in lean mice. Conversely, lean mice-Exos lead to a normalization of glucose tolerance improving systemic insulin sensitivity in obese mice. The authors suggested that ATMs influence metabolic events by paracrine signalling and that miR-155 contributes to the insulin resistant, glucose intolerant state downregulating PPARγ expression, a miR-155 target gene. [50].

Regarding neutrophils, they perform antimicrobial activity through myeloperoxidase and other proteins contained in granules. In obesity, neutrophil functions are altered with an oxygen free radical increase production, being already activated at the basal state with reduced ability to respond to the infectious stimulus [51]. Mature neutrophils will secrete pro-inflammatory cytokines and chemokines such as TNF-α, IL-1β, IL-8 and MIP-1α leading to the recruitment of other immune cells at the inflammation sites [52].

In the literature an increase of neutrophil amount in AT has been reported. In lean and obese mouse models, Talukdar et al. investigated the role of neutrophil elastase (NE), in the promotion of inflammatory response. The study confirmed the sustained increased in AT of neutrophils and suggested that NE could be a key effector in obesity related inflammation by recruiting different immune cells, and influencing their polarization state [53]. In addition, D’Abbondanza et al. compared the neutrophil extracellular trap (NET) levels, and their association with anthropometric and glycometabolic parameters, in healthy subjects and obese patients. Obesity resulted associated with increased neutrophil activation and NETs. Moreover, a higher blood pressure values and a worse glycol-metabolic profile were observed in patients than in controls. The concentration of MPO-DNA complexes was significantly associated to weight, body mass index (BMI), systolic and diastolic blood pressure, and glycol-metabolic profile [54].

Dendritic cells (DC) are considered the sentinels of the immune system. As antigen-presenting cells, they can stimulate the differentiation of T lymphocytes into pro-inflammatory Th1 cells or immunomodulatory Th2 cells [55,56]. DC are defined as CD11c positive cells and their maturation is modulated by leptin through anti-apoptotic nuclear factor-κB activation. [57]. As a result, leptin increases the secretion of DC pro-inflammatory cytokines such as IL-12, IL-6, and IL-1β and decreases anti-inflammatory IL-10 production.

In obesity, it has been reported that the recruitment of DC at the AT level (where they are not normally present) contribute to the establishment of a chronic inflammatory state [58]. In particular, Bertola et al. described the presence of a CD11c+/CD1c+ DC subset that correlated with the BMI and the Th17 increase in obese patients. This DC subset has been considered as an important regulator of AT inflammation, promoting the switch toward Th17 response in obesity-associated insulin resistance.

Different studies showed that DC infiltrated in AT may influence the balance Treg/Th17. In an obese leptin-deficient mouse model, Moraes-Vieira et al. have described that DC were able to regulate the inflammatory response modulating the Treg proliferation and differentiation [59]. Recently, Park et al. observed that DCs induced to maturate by IL-33, inhibited CD4+ T differentiation into Treg by decreasing Foxp3 expression [60].

NK cells are a subset of cytotoxic lymphocytes exerting their cytolytic activity by producing molecules such as granzymes and perforins. Once activated, they can produce pro- or anti-inflammatory cytokines (INF-γ, IL-6, TNF-α and IL-10), growth factors (GM-CSF; G-CSF) and chemokines (CCL-2 and IL-8). By producing INF-γ, NK are reported to regulate inflammation stimulating DC activation and M1 macrophages differentiation [61].

In obese patients visceral AT (VAT), local proliferation of NK has been reported, due to an increased expression of NK activating receptor on adipocytes [62]. Different studies reported an increase of NK in peripheral blood and/or in AT of obese and T2DM patients compared to healthy subjects [63,64,65]. Wensveen et al. found that a phenotypically distinct population of tissue-resident NK represented a crucial link between obesity-induced adipose stress and VAT inflammation [66].

In a mouse model, Lee et al. showed that NK regulate AT macrophages to promote insulin resistance in obesity. HFD also significantly improved the frequencies and the amount of NK cells in epididymal fat. It was shown that both sorted ATMs and epididymal fat expressed high amount of IL-15, which is important for NK cell proliferation/activation. The authors demonstrated that NK play crucial roles in the metabolic derangements associated with obesity. Moreover, NK depletion improved insulin resistance in liver and muscle. It has also suggested that obesity increased NK numbers and their activation in epididymal fat. In particular, NK cell depletion suppressed the expression of pro-inflammatory genes (e.g., Tnf and Itgax) and promoted the expression of anti-inflammatory or M2 genes (e.g., Il10 and Arg1) in sorted ATMs [67].

### 3.2. Adipokine Immunological Properties

AT plays a fundamental role in the regulation of glucose and lipid homeostasis. In healthy subjects, this tissue acts as an endocrine organ implicated in the production of several hormones such as leptin, adiponectin, resistin and other cytokines [68]. In a dysregulated setting, these molecules defined “adipokines” may induce inflammatory state with subsequent development of metabolic complications.

Leptin is a polypeptide of 167-amino acids, produced predominantly in the AT but also expressed in a variety of other tissues, including placenta, ovaries, mammary epithelium, bone marrow, and lymphoid tissues. Leptin modifies the action of insulin in isolated adipocytes [69] and stimulates glucose and fatty acid oxidation and lipolysis [70]. In obese and lean individuals, it is secreted following a similar pulsatile pattern, but in obesity, it is proven to have higher pulse amplitudes; therefore, obese children have higher leptin plasma levels, compared with normal-weight subjects. Leptin has pleiotropic activities. It acts as an endocrine signal by reducing appetite and stimulating energy expenditure. The increase in leptin levels in obese patients is associated with the state of leptin resistance [71]. Thus, leptin acts on the immune system by exerting a pro-inflammatory role. It binds receptors present on monocytes, macrophages, neutrophils, dendritic cells, NK cells, B and T lymphocytes [72] promoting a pro-inflammatory phenotypes and pro-inflammatory cytokine secretion, activating chemotaxis, production of reactive oxygen species (macrophages, neutrophils), cytotoxic activity and phagocytosis. It increases NK cytotoxic activity [73], stimulating secretion of pro-inflammatory cytokines such as IL-6 and TNF-α.

In adaptive immunity, leptin promotes the CD4+ T lymphocytes proliferation and differentiation toward a pro-inflammatory Th1 phenotype, increasing the production of pro-inflammatory cytokines such as INF-γ and IL-2 along with a decrease in the secretion of anti-inflammatory Th2 cytokines such as IL-10 and IL-4 [47,73,74]. It promotes the Th17 proliferation and decreases Treg expansion [47]. Leptin also increases the proliferation of B lymphocytes [30].

Adiponectin is the most abundant plasma adipokine secreted by healthy AT [75]. It is a hormone encoded by the ADIPOQ gene. Adiponectin is an anorexigenic peptide that regulates glucose uptake and fatty acid breakdown in skeletal muscle and AT.

This hormone enhances insulin sensitivity through increased fatty acid oxidation and inhibition of hepatic glucose production.

Adiponectin also exerts immunomodulatory effects, reducing the secretion of pro-inflammatory cytokines such as TNF-α, IL-6, monocyte chemoattractant protein 1 (MCP1-CCL2) and increasing anti-inflammatory cytokines (like IL-10) production [76] by macrophages.

Plasma adiponectin levels are found to be lower in obese than lean subjects. Moreover, remarkable negative correlations between plasma adiponectin levels and BMI have been shown both in humans and animals [77,78,79]. This condition leads to an increased production of pro-inflammatory cytokines such as TNF-α and IL-6 [80].

Resistin is an adipokine also known as AT-specific secretory factor (ADSF), primitively discovered in mice and named for its ability to interfere (resist) with insulin action [81]. It is a cysteine-rich peptide hormone derived from AT encoded by the RETN gene, that is expressed in several cell types including adipocytes and mononuclear white blood cells. It has been described in humans that resistin alone can promote inflammation [82,83] and although its role needs to be elucidated, it seems to be involved in a pathway inducing the differentiation towards pro-inflammatory macrophages [84]. It is known that resistin suppresses the ability of insulin to stimulate cellular glucose uptake, playing a role in obesity, insulin resistance and diabetes.

## 4. CD4+ T Cell Subpopulations

CD4+ T lymphocytes represent a functionally heterogeneous cell subpopulation [85]. CD4+ cells could be divided into different T helper (Th) subsets based on: (i) the expression of surface markers, (ii) the type of secreted cytokines and iii) the different cellular targets. In general, the main role of CD4+ cells is to activate, enhance and regulate the action of other immune system cells. Th cells include effector cells, such as Th1, Th2 and Th17, that protect from pathogens, and regulatory T cells (Treg) that can inhibit the effector cells in same particular conditions such as autoimmune responses.

The CD4+ cell differentiation into the different subpopulations is driven by the microenvironment in which the activated antigen-presenting cell resides [86]. In particular, T cell maturation is promoted by lineage-specific transcriptional factors that regulate the expression of specific surface receptors and the secretion of pro- or anti-inflammatory cytokines (Figure 2).

In the presence of an inflammatory state, such as the obesity-dependent inflammation present in AT, different studies have observed an increase of circulating Th17 and Treg [87] and a correlation with dysregulation of CD4+ subset [88]. As describe above, obesity is actually considered a pathological condition, characterized by a chronic low-grade inflammation in AT. The local obesity related inflammation is principally due to the release of pro-inflammatory cytokines, chemokines and adipokines (IL-6, IFN-γ, TNF-α, IL-1β, RANTES, MCP-1 and SDF-1α) which in turn promote the recruitment of immune cells into the AT. The infiltrating immune cells include macrophages M1 and M2, NK and different subtype of CD8+ and CD4+ cells [89]. In particular, it has been reported an increased number of CD4+ cells and a diminished amount of Treg [87]. The protective effect of Treg in obesity and in metabolic dysfunctions has been widely explored [90,91,92]. It is assumed that in obese patients, lipotoxicity has a fundamental role and leads to the Treg/Th17 imbalance of cells and to the development of obesity-related T2DM.

### 4.1. T Helpher 17 Lymphocytes (Th17)

The differentiation of naive T lymphocytes toward Th17 is driven by several cytokines secreted by antigen-presenting cells (APCs), including IL-6, IL-1-β, IL-21, TGF-β and IL-23 that activate the expression of the lineage specific transcription factor RORγt [93]. It was assumed that during the RORγt expression, autocrine and paracrine TGF-β acts in synergy with IL-6 amplifying the Th17 maturation and with IL-21 enhancing Th17 differentiation [94]. Moreover, TGF-β induces the surface expression of IL-23 receptor on differentiated Th17 making cells responsive to IL-23 action, that plays a key role in the differentiation, expansion and maintenance of the Th17 [95].

The maturation process requires the activation of the T cell specific surface receptors CD28 and CTLA-4. Recently, the presence of the ligand ICOS (CD278), an inducible co-stimulatory molecule important for the efficient development of normal and pathological immune reactions, on activated T cell surfaces has been also described. Th17, subset of CD4+ effector cells, play a role in adaptive immunity, contrasting infections caused by extracellular pathogens (fungi and bacteria). Of interest, in association with the soluble factor CXCL-13, they closely interact also with B cells recruited at the site of infection.

Th17 are characterized by the production of IL-17, specifically the two isoforms A and F [96]. IL-17A and IL-17F, binding to IL-17 receptor on epithelial and innate immunity cells, stimulate the production of G-CSF and IL-8 (CXCL-8), leading to neutrophil recruitment. Moreover, during their expansion in the site of inflammation, IL-17 can stimulate the release of several other pro-inflammatory molecules, such as IL-6, IL-21, IL-22, chemokines, metalloproteases (MMPs), TNF-α and GM-CSF, inducing the IL-23 receptor expression on Th17 surface [97].

The involvement of Th17 in the induction and progression of several inflammatory and autoimmune diseases has been demonstrated [98]. For example, it has been shown that the IL-17A, as marker of Th17, plays a role in the onset of atherosclerosis, which is a complication of the obese state [99,100].

### 4.2. Regulatory T Lymphocytes (Treg)

Treg play an essential role in the regulation of the immune response. They act modulating effector T and B cells in order to maintain proper immune homeostasis [101]. It has been reported that Treg lymphocytes promote tolerance to self-antigens and prevent the onset of autoimmune diseases and allergies [102].

Treg are responsible for the secretion of several inhibitory cytokines, such as TGF-β, IL-10 and IL-35 and the production of granzyme B. It has been described that IL-10 exert an important immunosuppressive action blocking the release of pro-inflammatory cytokines and the co-stimulation by CD28 and ICOS [103]. Additionally, granzyme B may induce effector cell apoptosis and may exert immunosuppressive effect through interactions with two inhibitory receptors express on Treg: lymphocyte-activation gene function 3 (LAG3) and TIGIT receptors [104]. Moreover, granzyme B blocks the co-stimulation necessary for the activation of effector T cells [105].

Treg differentiation is driven by TCR signalling, in presence of TGF-β, IL-2 and other costimulatory molecules. Mature Treg are defined by the expression of CD4+ (T helper lymphocytes), high levels of CD25 (interleukin IL-2 receptor), low levels of CD127 (interleukin IL-7 receptor) and high levels of the forkhead box P3 (FoxP3, suppressor of IL-2 transcription) [106].

It is well known that Treg levels increase in the presence of an inflammatory state in different tissues. In AT, Treg prevent metabolic disorders by a direct interaction with macrophages, reducing the local inflammation. In particular, it has been reported that in AT macrophages inhibit T effector cell activation acting on APCs. Moreover, Treg and pro-inflammatory macrophages antagonize each other functions [107,108]. As result, the imbalance of the number and the function of Treg/macrophage has been considered a crucial factor in obesity with IR. In line with these observations, Zhong et al. described a previously unrecognized homeostatic role for CD80 and CD86, costimulatory B7 molecules, which may reduce adipose inflammation by maintaining Treg numbers in AT. In humans and mouse model, CD80 and CD86 levels were negatively correlated with the degree of IR and the infiltration of macrophages in AT. In obese condition, reduction of B7 expression appeared to directly decrease Treg proliferation and function, leading to excessive amount of pro-inflammatory macrophages and the development of IR. CD80/CD86 double knockout (KO) mice had enhanced adipose macrophage inflammation and IR under both high-fat and normal diet conditions, accompanied by reduced Treg development and proliferation. Interesting, the authors reported that an adoptive transfer of Treg could reversed IR and adipose inflammation in double KO mice [109].

### 4.3. Treg/Th17 Balance

Recently, a mechanism for the maintenance of Treg/Th17 balance during immune response, thanks to cytokines that influence the differentiation of one subpopulation and antagonize the development of the other cell type, has been hypothesized [110]. For example: (i) TGF-β supports the survival and function of Th17 and Treg, (ii) IL-2, important for Treg proliferation and function, inhibits the development of Th17; (iii) IL-21, which plays a role in the differentiation of Th17, suppresses the generation of Treg [88]. Moreover, it has been reported that FoxP3, expressed by Treg, and RORγt, present on Th17, inhibit each other’s functions [98]. Based on these observations, it has been hypothesized that the production of Treg and Th17 are inversely correlated.

The balance between Treg and Th17 is important in regulating the inflammatory response, in particular, an increase of Th17 and/or a decrease of Treg can cause local and systemic inflammation or autoimmune disease. In line with this observation, it has been reported a correlation between elevated IL-6 level and the presence of elevated number of Th17. In fact, IL-6 could be an important factor in the activation of the subset differentiation, while Foxp3 can directly interact with RORγt and inhibits Th17 differentiation, influencing the Treg/Th17 balance [111].

Altered Treg quality and quantity have been described in autoimmune disease such as type I diabetes, multiple sclerosis, systemic lupus erythematosus, rheumatoid arthritis [112,113,114]. The importance of Treg/Th17 balance in primary biliary cirrhosis has been demonstrated in mouse model knockout for CD25. The loss of Treg function leading to an increase of Th17, caused the development of autoantibodies [113]. Th17 dysregulation has been described in different models of autoimmune diseases such as chronic colitis [115], autoimmune encephalitis [116] and psoriasis [117].

In AT, several efforts have been performed to understand the maintenance of Treg/Th17 balance. Fabrizi et al. investigated the possible role of IL-21 in an IL-21 knockout (KO) mouse model and, in parallel, in AT from subcutaneous and visceral depots [118]. The finding of a significant increase in the number of Treg in animal model and the correlation between IL-21R and TNF-α both in visceral and in subcutaneous AT, indicated a possible involvement of IL-21 signalling in the development of T cell subset dysregulation. All these evaluations led to the hypothesis that IL-21 exerts negative regulation on Treg activity, favoring the development and maintenance of the obesity-induced inflammatory state.

## 5. Th17 and Treg Dysregulation in Obesity-Induced Inflammation

The relationship between immunity and metabolism plays an important role in obesity and related diseases. Indeed, there are evidences of an increase in inflammatory indices and alterations in immune homeostasis [8,9]. It is well known that the VAT presents higher levels of acute phase proteins compared to the subcutaneous one. Moreover, due to its vascularization, VAT is characterized by the presence of several immune cells, representing the source of the pro-inflammatory cytokines traced in different studies [119]. Specifically, it was demonstrated the presence of inflammatory macrophages M2, secreting large amount of pro-inflammatory chemotactic proteins, such as IL-12 and IL-1β, in VAT of obese subjects [87]. It has been reported that in obese subjects high levels of pro-inflammatory cytokines positively correlate with the development of IR and the onset of T2DM [120]. In addition, other soluble factors such as Irsin, OX40 and IGF1, are described in obesity and diabetic-related complications [121,122,123]. In the last decades, several studies defined a fundamental role of Th17 and Treg imbalance in obesity-dependent inflammation. The close relationship between obesity and Th17 and Treg has been firstly demonstrated in animal models, where Th17 increase and Treg reduction was observed in VAT [124]. The finding that the imbalance of Treg/Th17 can mediate the occurrence of obesity-related inflammation and metabolic disorder has been also supported by a study in obese mice where the overexpression of protein tyrosine phosphatase N2 (PTPN2) was suggested to inhibit the differentiation of Th17 while promoting Treg differentiation [125]. Interestingly, in a diet-induced obese C57BL/6 mouse model, Jhun et al. demonstrated that genes associated with retinoid-interferon-induced mortality 19 (GRIM19) could attenuate obesity. The results showed that in GRIM+ transgenic mice, IL-17 and pSTAT3 levels were down-regulated, while Treg and pSTAT5 expression was up-regulated. The conclusions of this study suggested that the inflammatory state obesity-dependent can be counteract by regulating Treg/Th17 balance through the suppression of STAT3 and the induction of STAT5 [126]. Furthermore it is known that lipotoxicity alters the Treg/Th17 ratio not only in AT but also in the gut and liver [127,128]. Moreover in obese subjects, it is shown that an IL-17 decrease promotes adipogenesis [93]. Th17 cells connect innate and adaptive immunity and play multiple roles: protective at the level of mucous membranes and pro-inflammatory in several inflammatory diseases including obesity [129,130,131]. Indeed, Th1 cells play a key role in the development of obesity by producing IFN-γ both in adult and children population [132,133]. Several studies have shown that the total number of CD3+, CD4+, and CD8+ T cells is increased in the AT of obese subjects; in addition, there appears to be a positive correlation between adiposity levels, BMI, and total T-cell numbers [134,135]. This condition correlates with a switch of pro-inflammatory phenotype of CD4+ cells and a decrease in peripheral T lymphocytes. In addition, leptin promotes T-cell proliferation and Th1/Th17 cytokine secretion such as IFN-γ, and prevents their apoptosis through the mTOR signalling pathway after antigen stimulation [136,137], maintaining chronic, obesity-related low-grade inflammation. Treg act positively by suppressing inflammation through the production of immunosuppressive cytokines such as IL-10 and TGF-β, which appear to regulate the differentiation and proliferation of lymphocytes and other immune cells in a manner that promotes immune tolerance [13]. Th17, on the other hand, produce several types of pro-inflammatory cytokines such as IL-17A, IL-17F, IL-21, and IL-22 [138]. Therefore, it is possible to say that the two subsets play opposite roles. Studies in the literature have remarkably demonstrated that in healthy conditions their percentage is kept in balance, but this proportion appears to be altered in pathological condition such as obesity, which involves a marked Treg reduction and an increase in Th17 in AT [139]. In addition, high frequency of Th17 and high level of RORC2 are reported in VAT of obese patients with metabolic abnormalities [139]. Very recently, Vega-Cardenas et al. defined an association between RORC2 and miR-326, one of the critical miRNAs involved in the Th17 production and IL-17A secretion [140]. However, not all studies are concordant today [141,142]. Wen J et al. showed that the percentage of Treg and the Treg/Th17 ratio tend to decrease in overweight, obese and dysmetabolic patients [17]. At the intestinal level, healthy microflora induces and promotes the development of non-pathogenic Th17 that enhance the protection of mucous membranes from pathogens [143]. However, obesity leads to the disruption of the homeostasis of the intestinal microflora, promoting the development of pathogenic flora [144]. Hypercaloric diets cause a reduction in IL-17 production in the gut and a switch to the “pathogenic” Th17 phenotype [93]. In conclusion, the excess of weight alters immune and metabolic homeostasis. The development of systemic inflammation induces reprogramming of Th17 that acquire new properties. The heterogeneity of Th17 cells, associated with changes in microenvironmental factors, contributes to the development of severe inflammatory/autoimmune disorders [145].

The interaction of gut microbiome with immune cells and adipose tissue is also considered an additional mechanism responsible for inflammatory condition in childhood obesity [146]. The impact of gut microbiota have been extensively studied in order to understand the host-gut microbiota interactions. The human digestive tract microbiota influences the integrity of the intestinal barrier. In healthy subjects, the gut microbiota is in homeostasis. The alteration of intestinal microbiota is reported to be linked to the development of inflammation present in different pathological conditions such as chronic intestinal disease, inflammatory bowel disease, asthma, diabetes mellitus, and metabolic syndrome [147,148,149,150,151,152]. The dysregulation in nutrient absorption and metabolism has been associated to obesity [153,154]. It has been described that obese patients have a reduced diversity of intestinal bacteria compared to healthy subjects [155], with lower levels of Bacteroides, known as lean microbiota, and higher levels of Firmicutes, called obese microbiota, compared to normal weight subjects [156]. It has been reported that gut microbiome interacts with immune cells and adipocytes and at the same time, immune cells and adipocytes regulate its functions. Recently, in HDF-fed obese people, Luck et al. showed that the intestinal dysbiosis promote the secretion of gamma IFN-γ by T cells and the reduction of Tregs, Th17 and IL-22 reducing homeostasis [157].

## 6. Treg/Th17 Dysregulation and Gluco-Metabolic Abnormalities

It is well known that metabolic reprogramming is critically important for lymphocytes. Manipulating metabolic pathways can shape the differentiation and function of these cells [158]. Metabolism furnishes T cells with energy and precursors for many biological processes. Some primary metabolic pathways, such as oxidative phosphorylation, fatty acid oxidation and glycolysis, are considered to play fundamental roles in T cell activation and differentiation.

Functional IL-6 and TGF-β signalling are the initial events needed to start Th17 differentiation. IL-23 and IL-21 play a fundamental role in the maintenance of the Th17 progeny by increasing the transcription of IL-17 and other cytokines. STAT3 (signal transducer and activator of transcription 3), which is critical for the effects of IL-6, IL-21, and IL-23 is required for Th17 differentiation while, on the other hand, the same cytokines are fundamental to initiate the signalling pathways [159,160]. Th17 were identified as a new progeny of CD4+ T helper cells after the finding that experimental autoimmune encephalomyelitis in animal models was caused by high levels of IL-23 rather than IL-12 and Th1 cells [161]. Consequently, it became evident that the function of IL-23 was to promote differentiation and proliferation of the IL-17 secreting cells, classified as Th17. Nevertheless, IL-23 alone has been found to be unable to make naïve T cells differentiate into Th17; some studies showed that polarization of Th17 can be appropriately induced by IL-6 and TGF-β1, which activate STAT3 and Smad family proteins, respectively [162,163]. Dormant naive T cells have relatively small energetic demands, generally supported by glucose oxidation via the Krebs cycle and the oxidation of lipids with low levels of glycolysis, in order to maintain cellular homeostasis [164]. After stimulation, they start to proliferate and differentiate into Th cells. This requires metabolic reprogramming to support their rapid expansion and further functions, such as synthesis of macromolecules, intracellular mediators and cytokines. First, glucose transporter (GLUT) and alanine–serine–cysteine transporter (ASCT2) become highly expressed, then glycolysis fatty acid metabolism, along with OXPHOS, PPP, hexosamine pathway all become active [165].

Aerobic glycolysis and glutamine catabolism become the main pathways, along with a down-regulation of the metabolic characteristic processes of resting cells. The T-cell receptor starts the signalling cascade, along with MAPK (mitogen-activated proteinkinase) ERK (extracellular signal-regulated kinase), PI3K (phosphoinositol-3 kinase), mTOR (mammalian target of rapamycin) and NfκB (nuclear factor-κB). These costimulatory molecules are necessary to induce the Myc and HIF-1α transcription factors, known to induce various gene expression implicated in glycolysis and glutaminolysis [166,167]. If this upregulation of glucose metabolism is not achieved, T-cell differentiation, both in vitro and in vivo, is inhibited [168].

Cellular crosstalk plays a critical role in regulating T-helper maturation and differentiation. As described above, specific cytokines, for the so-called antagonism effect, while drive the generation and function of the specific subset, work to reduce alternative pathways [169].

Treg/Th17 imbalance has been associated with metabolic dysregulation in diabetic patients. A study conducted in HDF-fed transgenic animal models revealed that Treg expansion determined a significant reduction in blood glucose, insulin resistance, and increase in glucose tolerance [93]. In addition, different researches showed a Treg decrease in VAT or in peripheral blood derived from obese and diabetic adults [170,171]. In a recent study, Wen et al. observed Treg/Th17 imbalance in obese and overweight subjects with or without metabolic dysfunctions. The authors reported a severe decrease in Treg/Th17 ratio in peripheral blood of overweight/obese patients with impaired glucose regulation or T2DM compared to healthy subjects or overweight/obese patients with normal glucose tolerance. Moreover, the authors observed that the degree of the imbalance was positively correlated with the exacerbation of metabolic alterations. Furthermore, the serum IL-6 level in patients with metabolic compliance was higher than in controls while the Treg/Th17 ratio was negatively correlated with HbA1c [17].

In the last decades, several studies showed that functional defects of Treg are correlated with the development of IR [172,173,174] (Figure 3). In fact, the IR is known to be linked to the promotion of T cell activation in obese subjects [175]. Recently, Gilleron et al. observed that adipocyte hypertrophy and IR in obese mice were driven by an increase in adipose Th17 and a decrease in adipose Treg. In particular, it has been also described that Treg/Th17 imbalance reduced adipogenesis [176]. In another study, the effects of OX40 has been associated to Th cell differentiation, proliferation and reduction of Treg regulatory activity. The authors underlined how Treg/Th17 balance was crucial for the development of AT inflammation and IR [122]. A lot of studies have shown that pro-inflammatory cytokines such as IL-6, IL-1β, TNF-α, NF-κB can lead to IR and consequently to the development of related diseases like metabolic syndrome and/or diabetes [177,178].

Inflammation activates macrophages, along with the release and activation of inflammatory molecules. TNF-α also induces MCP-1 production, which ultimately activates the chemotactic migration of macrophages. These mechanisms result in the inhibition of insulin signalling and sensitivity. Inflammatory molecules also increase peripheral free fatty acid levels through lipolysis, which further aggravates insulin resistance. This determines a perpetual cycle, in which altered values of blood sugar and lipids lead to many related complications [179]. The specific therapeutic treatments for obesity and/or metabolic dysfunctions have been implicated in the regulation of Th17 and Treg balance. Of interest, Martinez-Sanchez et al. showed that high levels of insulin increased the differentiation toward Th17 and decreased Treg maturation in vitro, supporting the hypothesis that Treg/Th17 imbalance could be a mechanism for the onset of metabolic disorders in obesity [124]. Moreover, several data suggest that high insulin levels in obesity cause an inflammatory state by impairing Treg-induced suppression. It has been demonstrated that insulin affects Treg receptors, decreasing IL-10 release through activation of the AKT protein signalling pathway and mTOR [180].

Metformin is a recommended drug for T2DM treatment that improves insulin sensitivity and prevents hyperglycemia by reducing chronic inflammation. It has been reported that the treatment of T2DM patients with metformin induced a decrease in Th17 [181,182,183]. Similarly, in a recent study, Borzouei et al. investigated the expression of immune factors related to Th17 such as RORγt, STAT3, and IL-17, and Treg such as FoxP3, STAT5, and IL-10 in T2DM patients before and after empagliflozin plus metformin and gliclazide. After six months of treatment, a significantly reduction in RORγt and a significantly increase in FoxP3 and STAT5 were reported; IL-17 level was decreased while IL-10 level was enhanced compared to patients treated with only metformin and gliclazide. Empagliflozin showed anti-proliferative and anti-inflammatory effects reducing Th17 and increasing at the same time Treg levels [184].

Han et al. investigated the possible influence of IL-33 on Treg in VAT of four-week-old male mice. They observed that in HFD obese mice, Treg levels diminished but the treatment with IL-33 reversed this condition and counteracted VAT inflammation, leading to a reduction in hyperinsulinemia and IR [185].

Recently, the effect of different molecules on Treg/Th17 ratio has been investigated in animal models. For example, Wei et al. described a possible amelioration in obesity-dependent IR by acacetin in a mouse model. Acacetin seems to down-regulate IL-17 and up-regulate Foxp3 expression, promoting Treg/Th17 balance via targeting miR-23b-3p/NEU1 axis [186]. Finally, a therapeutic effect of epigallocatechin-3-gallate, has been observed in obese mice, showing a significant reduction in weight, LDL-cholesterol and triglyceride levels. Moreover, a higher Treg/Th17 ratio was reported [187].

Recently, data investigating the role of Th17 and Treg in metabolic derangement in pediatric patients are reported. Calcaterra et al. [16] evaluated the Treg/Th17 balance in obese children, in relation with their metabolic status. A correlation between Th17 and systolic hypertension, Treg/Th17 ratio and HOMA-IR was noted. The Treg/Th17 balance appeared to be involved in glycemic homeostasis and blood pressure control [16]. In pediatric patients with chronic inflammation associated-obesity, it has been reported a Th17 involvement, evaluating the frequency of this T cell subset in the peripheral blood. Children with central obesity were characterized by higher percentages of Th17 compared to normal weight children. Moreover, Treg/Th17 ratio positively correlated with total plasma cholesterol concentration [188]. Still referring to paediatric patients, Schindler et al. observed a correlation between overweight and elevated frequency of circulating Th17, IL-17A mRNA levels and RORC. Moreover, Th17 frequency positively correlated with BMI [189]. Of interest, in contrast to previous studies reporting elevated IL-17 levels in obese adults, Jung et al. observed a significant decrease in IL-17 levels in overweight adolescents compared to lean controls. The authors suggested as a possible explanation that the disease conditions associated with obesity such as hypertension and vascular pathologies were not yet present in overweight teenagers [190].

## 7. Conclusions

Children and adolescents with obesity have a high risk of developing impaired glucose metabolism. Adipose tissue appears to be involved in T cell regulation of tissue inflammatory and in Treg/Th17 imbalance influencing metabolic responses. In the patient care, immune monitoring could play an important role to define preventive strategies of pediatric metabolic disease treatments.

## Figures and Tables

**Figure 1 children-08-00554-f001:**
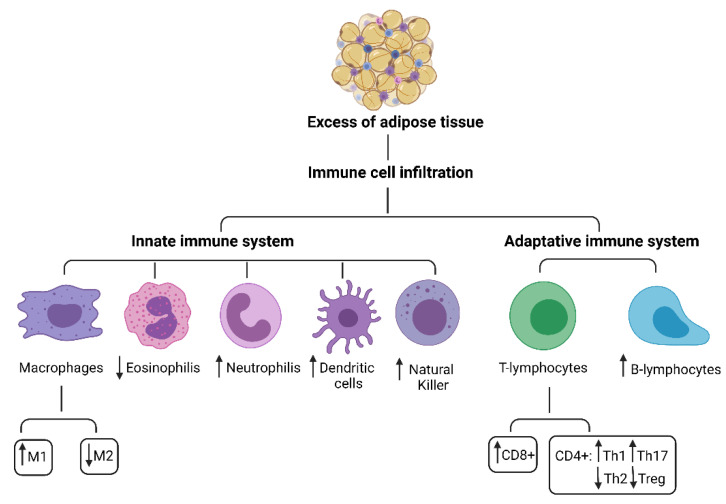
Innate and adaptive immune system cells in adipose tissue.

**Figure 2 children-08-00554-f002:**
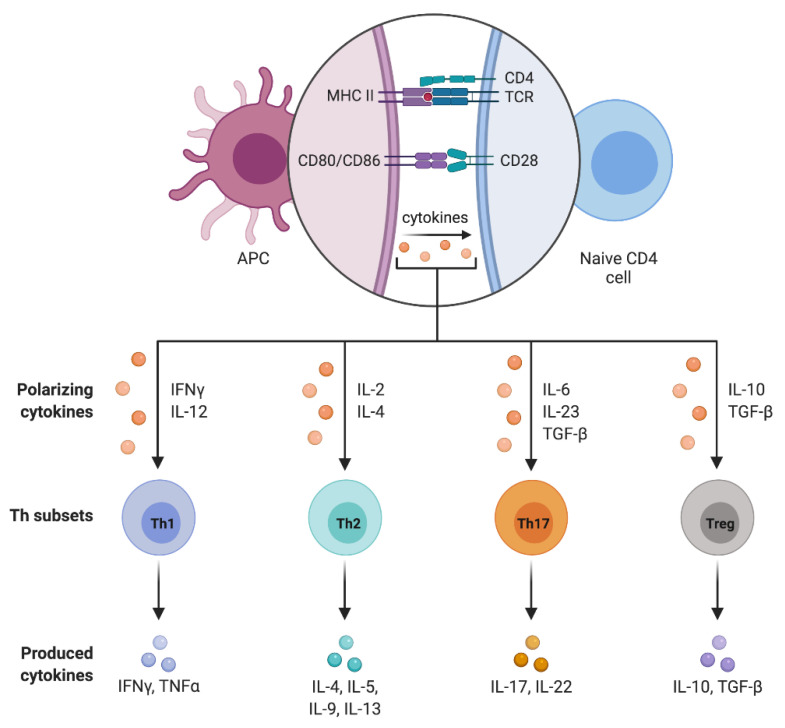
CD4+ T cell subpopulations.

**Figure 3 children-08-00554-f003:**
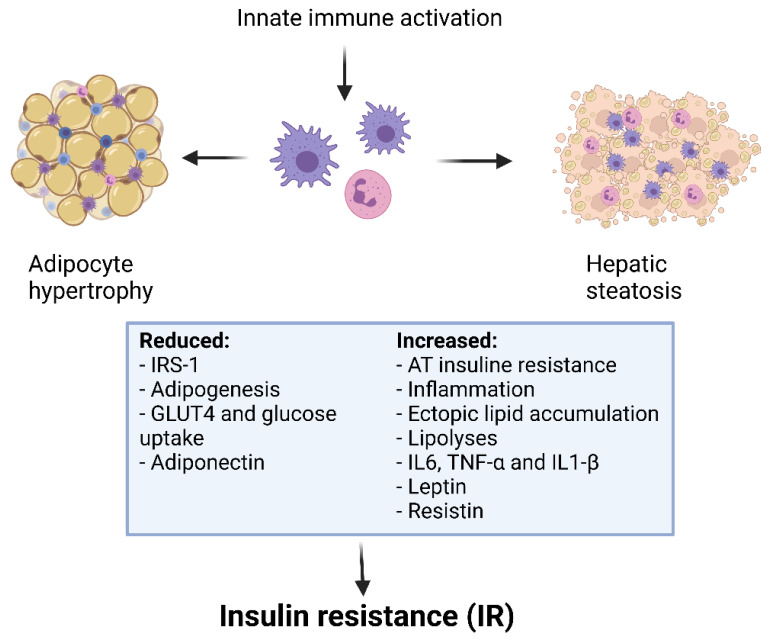
Obesity, inflammation and insulin resistance.

## Data Availability

The data presented in this study are available on request from the corresponding author.

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
