# Peer review of "Adipose Tissue Immunomodulation and Treg/Th17 Imbalance in the Impaired Glucose Metabolism of Children with Obesity"

_children, 2021, doi:10.3390/children8070554_

Round 1
Reviewer 1 Report
The narrative review by Croce et al. addresses the role of adipose tissue immunomodulation and the potential role played by Treg/Th17 imbalance on the impaired glucose metabolism in pediatric obesity. This narrative literature review describes and discusses adipose tissue immunomodulation from a theoretical and contextual point of view. Unfortunately, this type of review (narrative) does not specify the precise methodological approaches used to conduct the review nor the evaluation criteria for inclusion of retrieved articles during databases search. A systemic literature review would have been the preferable choice to answer their specific research questions since they are conducted using rigorous methodological approaches to identify, select, and critically evaluate results of the studies included in the literature review.
Still, the authors in this narrative review do succeed in presenting a critical analysis of the pertinent, up-to-date literature published in scientific journals (in the English language).
Methods section: Consider adding a more detailed description of the number of publications reviewed and how studies were chosen (inclusion/exclusion criteria of studies).
The figures (1-3) are exceptional. They enable the reader to visualize the cells and sub-populations.
There are some typos throughout the text such as in the legend to figure 3 , "inflammationa" should be "inflammation". Please review the English and the layout of paragraphs, at times only one sentence per paragraph.
Author Response
REVIEWER 1
The narrative review by Croce et al. addresses the role of adipose tissue immunomodulation and the potential role played by Treg/Th17 imbalance on the impaired glucose metabolism in pediatric obesity. This narrative literature review describes and discusses adipose tissue immunomodulation from a theoretical and contextual point of view. Unfortunately, this type of review (narrative) does not specify the precise methodological approaches used to conduct the review nor the evaluation criteria for inclusion of retrieved articles during databases search. A systemic literature review would have been the preferable choice to answer their specific research questions since they are conducted using rigorous methodological approaches to identify, select, and critically evaluate results of the studies included in the literature review.
Still, the authors in this narrative review do succeed in presenting a critical analysis of the pertinent, up-to-date literature published in scientific journals (in the English language).
R: Thank you for your comments. We aggree with the reviewer on the limits of the narrative review. We specified in the methods that as a narrative review, several statements based on expert opinions and not evidence-based or supported by appropriate in vitro or in vivo studies were included.
Methods section: Consider adding a more detailed description of the number of publications reviewed and how studies were chosen (inclusion/exclusion criteria of studies).
R: As required, we added additional information
The figures (1-3) are exceptional. They enable the reader to visualize the cells and sub-populations.
R: Thank you for your positive comment
There are some typos throughout the text such as in the legend to figure 3 , "inflammationa" should be "inflammation". Please review the English and the layout of paragraphs, at times only one sentence per paragraph.
R: We revised the English form

Reviewer 2 Report
The review article entitled " Adipose tissue immunomodulation and Treg/Th17 imbalance in the impaired glucose metabolism in children with obesity" explains the role of adipose tissue immunomodulation and the potential role played by Treg/T17 in pediatric obesity. The article has all the basic textbook material but is missing the critical gaps.
- The authors need to show the stats of pediatric obesity and its prevalence in the introduction.
- The article has too many short paragraphs which needs to be compiled to form one full paragraph.
- There is no need to have Methods section, this can ne the last paragraph of the introduction.
- Line 76 – 78 can be brought to author contribution section and be removed from the method section.
- The article lacks supportive clinical results published from other groups.
- The role of microorganism needs to be discussed, which plays a major role in low grade inflammation and stimulation of cytokines.
- Lines 139-145 does not fit in that section.
- The article needs major editing.
Author Response
REVIEWER 2
The review article entitled " Adipose tissue immunomodulation and Treg/Th17 imbalance in the impaired glucose metabolism in children with obesity" explains the role of adipose tissue immunomodulation and the potential role played by Treg/T17 in pediatric obesity. The article has all the basic textbook material but is missing the critical gaps.
- The authors need to show the stats of pediatric obesity and its prevalence in the introduction.
R: As required, we added additional information
- The article has too many short paragraphs which needs to be compiled to form one full paragraph.
R: we adapted the paragraph 3.
- There is no need to have Methods section, this can ne the last paragraph of the introduction.
R: We understand your comment, however, we had to meet the comment of the other reviewer, who asked for an expansion of the method section.
- Line 76 – 78 can be brought to author contribution section and be removed from the method section.
R: please see the response to point 3
- The article lacks supportive clinical results published from other groups.
R: Using the selected keywords we did not find other papers reporting clinical results other than those already included (references 188,189 and 190)
- The role of microorganism needs to be discussed, which plays a major role in low grade inflammation and stimulation of cytokines.
R: thank you for your comment, we included and discuss this point.
- Lines 139-145 does not fit in that section.
R: we agree with this observation and we moved lines 139-145 in paragraph 3.1
- The article needs major editing.
R: Thank you for this comment, we try to reediting the manuscript

Round 2
Reviewer 2 Report
None